# Creating and Reliving the Moment: Using Musical Improvisation and Care Aesthetics as a Lens of Connection and Self-Expression for Younger People Living with Dementia

**DOI:** 10.3390/ijerph21080972

**Published:** 2024-07-25

**Authors:** Robyn Dowlen, Dougal Henry James McPherson, Caroline Swarbrick, Lizzie Hoskin, James Thompson, John Keady

**Affiliations:** 1Division of Psychology and Mental Health, University of Manchester, Manchester M13 9PL, UK; 2Creative Manchester, University of Manchester, Manchester M13 9PL, UK; 3Division of Health Research, Lancaster University, Lancaster LA1 4YW, UK; 4Manchester Camerata, The Monastery, Manchester M12 5WF, UK; 5School of Arts, Languages and Cultures, University of Manchester, Manchester, M13 9PL, UK; 6Division of Nursing, Midwifery and Social Work, University of Manchester, Manchester M13 9PL, UK; 7Greater Manchester Mental Health NHS Foundation Trust, Prestwich, Manchester M25 3BL, UK

**Keywords:** young onset dementia, group music-making, ‘in the moment’, visual methods, care aesthetics, musical improvisation

## Abstract

Musical improvisation is a generative process of spontaneously creating music ‘in the moment’. For people with young onset dementia, musical improvisation provides an extended opportunity for creative self-expression and connection to one’s own body and life story. Using visual research methods, including video elicitation interviews, this paper explores the ‘in the moment’ musical experiences of five people living with young onset dementia who took part in a 15-week improvised music-making programme (Music in Mind). We frame the exploration of the group’s musical experiences through the emerging lens of ‘care aesthetics’—a concept that identifies the sensory relations and embodied practices between two (or more) people in a caring relationship. In the context of this analysis, we look to the caring practices by, with, and between people living with dementia, their family members, and the musicians who lead the programme and the relationship of these practices to feelings of self-expression and meaningful connection. Musical improvisation has the potential to support the psychological, social, and spiritual wellbeing of people living with young onset dementia. In applying a lens of care aesthetics, it is possible to observe the micro-level experiences of people living with dementia and their family carers.

## 1. Introduction

In a systematic review and meta-analysis of the global prevalence of young onset dementia (i.e., dementia occurring before the age of 65 years), Hendriks et al. [1] proposed that an estimated 3.9 million people live with the condition globally. In the United Kingdom (UK), and building on the earlier Dementia UK report [2], Carter [3] suggested that there are around 70,800 younger people living with dementia. Moreover, whilst dementia can be present across the lifespan, including in rare instances in children [4], the Dementia UK report [2] suggests that for younger people living with dementia, it is the 60–64-year-old age group that is most at risk of being diagnosed with the condition and that men have a slightly increased risk of being diagnosed; a positionality that is further increased in frontotemporal dementia [2,3]. No matter the diagnostic category, a diagnosis of dementia in mid-life can have serious and disruptive consequences for a person’s life choices and decision-making abilities. 

As studies have suggested, for people living with young onset dementia, this ‘off-time dependency’ [5] is likely to affect a wide range of intrinsic and extrinsic life factors including, but not limited to, changing relationships and roles within the family structure [6,7,8,9]; workforce and retirement planning issues [10]; leisure pursuits [11,12]; family finances due to the likelihood of mortgage payments and other outstanding loans and debts still needing to be repaid [5,13]; changes to personal wellbeing, self-identity, and self-confidence [13,14]; social stigma [15]; and receiving a delayed diagnosis due to the complexities involved [2,16]. Consequently, the health, wellbeing, and quality of life of younger people living with dementia and their support networks is likely to significantly worsen without access to timely and appropriate service support and improved professional awareness [14].

In the UK, the most recent National Institute for Health and Care Excellence (NICE) dementia guideline [17] revealed that the provision of evidence-based support for younger people living with dementia is lacking. Indeed, when young onset dementia is mentioned in the NICE guideline, it is either focussed on making professionals aware of the potential genetic causes of both Alzheimer’s disease and vascular dementia, or that professionals need to be aware of the ‘specific needs of younger people living with dementia’. However, these specific service support needs are not explicated within the guidance. To attempt to provide some clarity, Stamou et al. [18], in a recent survey of UK-based professionals (*n* = 233) involved in young onset dementia services, found the following three areas that best summarises these needs: first, for younger people living with dementia to understand the condition they live with and how to cope with the changes it brings; second, for younger people living with dementia to access age-appropriate services suitable for their social and physical/mental health needs; and third, for younger people living with dementia to maintain or regain a sense of connectedness and reciprocity within family and age-appropriate social contexts. 

In both the literature and everyday practices, one way for younger people living with dementia to regain this sense of connectedness is by taking part in cultural and arts-based activities. Studies have shown that younger people living with dementia have improved their overall sense of wellbeing by taking part in a range of activities including attending museums and galleries [19] and engaging in creative art and music-making [20,21]. As Nedlund et al. [22] and Dowlen and Fleetwood-Smith [23] have argued, engaging in these activities supports the everyday citizenship and human rights of (all) people living with dementia and positions them as active contributors to the places and spaces in which they live, rather than as passive recipients of care and support. In addition, a wealth of information can be gathered from examining creative experiences from an ‘in the moment’ perspective. For example, both family carers and healthcare professionals have reported valuing ‘in the moment’ musical experiences, noting that these do not lose their value even if the perceived benefits for the person living with dementia do not last beyond the music-making time [24]. 

One reported benefit of participation in arts and creativity is that a full and enjoyable immersion in the activity does not rely on biographical memory. This means that during such interactive participation, people living with dementia are not necessarily ‘judged’ or limited by the cognitive challenges that may affect them in other parts of their daily lives [25]. Engagement with the arts, especially those that are participatory in nature, affords people living with dementia meaningful opportunities to make things that are of, and for, ‘the moment’ through spontaneity, group co-creation [26], and improvisation [27]. Furthermore, Zeilig et al. [26] posited that by drawing on co-creative ‘moments’, it is possible to focus more extensively on creative, relational, and agential interactions between people with dementia, which supports creative agency. Drawing on a synthesis of a number of creative and arts-informed participatory studies conducted in dementia care, Keady et al. [28] defined ‘being in the moment’ in the following way: 


*Being in the moment is a relational, embodied, and multisensory human experience. It is both situational and autobiographical and can exist in a fleeting moment or for longer periods of time. All moments are considered to have personal significance, meaning and worth. *

*(p. 687.)*


Importantly, these authors [28] argued that ‘being in the moment’ is not an isolated event and that it exists within a continuum where moments are also created, ended, and (potentially) re-lived once again, thus creating a recurring cycle of moments. Viewing and valuing the lived experience of younger people living with dementia through such a lens offers the potential for the ripple effects of moments to be experienced by others in an interactive process. In this article, we will develop this linkage by exploring the role of musical improvisation, facilitated by professional musicians and music therapists, and how this provides moments of connection and opportunities for positive self-expression by people living with young onset dementia reimagined through a lens of care aesthetics. The following sections will provide an overview of musical improvisation and care aesthetics to provide context to this study. 

### 1.1. Improvisation

Improvisation is a generative process in which spontaneous expressions emerge in the moment. Whether enacted as a distinct mode of performance (such as in free jazz or contact improvisation), or applied within therapeutic or pedagogical contexts, it is dependent on ‘a myriad of constantly changing variables’ [29], which are located across social, as well as artistic, domains. In music, participants in a group improvisation co-create its rules and aesthetics, how it works and how it sounds, through ‘action and awareness that is situated, responsive, reflexive, and creative’ [30]; together, they generate rhythmic, harmonic, melodic, and textural material which invents its own way of proceeding as it unfolds [31]. Central to the development of disciplines such as music therapy [32,33], improvising methods have been engaged widely across the applied arts (within and beyond music), and a developing body of literature identifies the affordances of improvisation in fostering self- and group-creativity, as well as its positive impact for participant wellbeing [34,35,36]. Furthermore, improvisation removes the need to rely wholly on a shared musical lexicon which may be othering when not shared by all members of a group (for example, a reliance on a particular period or genre of music through reminiscence-based musical activities) [27]. 

Recent research demonstrates the diverse uses of improvisation in applied arts contexts, which range from supporting individuals with a personality disorder to develop confidence and trust [37], fostering self-belief with older adults living in community [38], supporting the rehabilitation and integration of refugees and asylum seekers [39], to understanding the capacities of scenic (theatrical) improvisation to facilitate wellness for young women of colour [40]. Group improvisation has been widely employed in teaching, facilitation, and therapeutic contexts as a means of creating an inclusive and accessible space for participation in collective creativity [29].

### 1.2. Care Aesthetics

Hanlon and Carlisle [41], in their writing on public health and health promotion, have argued for a ‘fifth wave in public health’ that avoids ‘reductionist studies that seek to show biomedical benefits from interventions that involve art and music’. They argue for approaches that are ‘integrative’ allowing us to experience what it is ‘to be fully human and live together in a healthy, sustainable, but also beautiful world’ [41]. They insist that our analyses of health-related practices should combine scientific understanding, an ethical orientation, and an appreciation of aesthetics. This focus is not unique for health researchers and in fact echoes the work of nursing academic Barbara Carper [42] who over 45 years ago argued for an attention to the ‘aesthetic pattern of knowing’ in nurse training. 

These perspectives have been drawn upon in the emerging concept of care aesthetics [43], which argues that care as a practice—whether in formal or informal health settings—has sensory, embodied, and crafted aspects that need to be acknowledged, and then following Hanlon and Carlisle [41], integrated with other modes of analysis. Rather than only noticing the technical or task-orientated aspects of care, care aesthetics seeks to draw attention to interactions between individuals, valuing the subtleties of micro movements, touches, glances, tones of voice, and body posture. The claim is that attention to these sensory practices, both in carers and in the responses of care recipients, helps reveal aspects of the qualities of a care experience that are too often overlooked. Moreover, as Thompson [43] suggested, one of the opportunities that care aesthetics provides is to focus on the ‘moment’ of care and to ensure that ‘caring with’ includes all parties directly or tangentially involved in the analysis.

Thus, by combining the concepts of musical improvisation and care aesthetics, and applying them to an existing data set on an improvised music-making for younger people living with dementia as conducted by the lead author and involved Manchester Camerata’s ‘Music in Mind’ programme [27,44,45,46,47], it allows a new research question to be explored which focusses on the salient sensory, relational, and embodied ‘caring moments’ and their transitional meaning. We will now attempt to answer this new research question in the following section, starting with an introduction to Music in Mind where the empirical component of this study is located before moving on to consider the original study design.

## 2. Materials and Methods

### 2.1. Music in Mind: An Introduction 

Manchester Camerata is a world-renowned chamber orchestra and charity based in Greater Manchester in the UK who deliver a wide range of community-based engagement work. As part of this delivery agenda, Music in Mind commenced in 2012 and is a 15–20-week, improvised, live, music-making programme for people living with dementia of all ages and their family carers, or those supporting them in a social or professional capacity. The Music in Mind programme can take place either in community or care home settings. 

Each Music in Mind session lasts about 1 hour and has (up to) 30 participants who sit in a circle alongside the musicians. Each session starts with a ‘Welcome Song’ and ends with a ‘Goodbye Song’ and is co-facilitated by a music therapist and specially trained orchestral musician from Manchester Camerata and is based on the principles underpinning musical improvisation; these include acceptance (all contributions are welcome), distributed creativity (all participants share equally in creating together), and a focus on being ‘in the moment’. The face-to-face delivery of Music in Mind centres choice and agency for the person living with dementia, enabling a democratisation of the musical space through supported improvisation using percussion instruments, the human voice, familiar objects (such as chairs, tables), and the body. 

Practically, group improvisation encourages participants to develop empathetic, interpersonal, and communicative skills through musical means. Co-creative interactions are foregrounded as each individual is responsible for the shape, content, and trajectory of a collectively created musical whole. Participants navigate the coming together of both musical and social identities, in a flexible arena of spontaneous play. The group responds ‘in the moment’ to changes in atmosphere and musical content sparked by its members’ musical contributions. The result is a nonreplicable, unpredictable, collectively generated emergent expression.

### 2.2. Study Participants

As shared in Table 1, the original study was conducted in 2017 and recruited five younger people living with dementia and three family carers to a 15-week Music in Mind programme delivered in a community setting in Greater Manchester. 

This study included people living with dementia who had capacity, as well as those who lacked capacity as determined by the UK’s Mental Capacity Act [48]. The capacity of people living with dementia was determined through consultation with family members and through ongoing monitoring using a process consent approach [49] which acknowledges ‘that ethical decisions and actions are context specific and centred on interdependence within a caring relationship and acknowledges that capacity is situational’.

This meant that the consent was assessed across the different situational contexts in the research (Music in Mind sessions vs. interview visit at home) and the role of the family carer in the consenting procedure was acknowledged throughout.

For the purposes of protecting the identities of those involved, all study participants have been given pseudonyms in line with the ethical permission and study protocol. The two Music in Mind practitioners who were delivering the project were also recruited as participants and are also referred to using pseudonyms.

### 2.3. Study Design and Analysis

The original study used a multiple-case study design [50] to gain an in-depth understanding of the ‘in the moment’ experiences of each younger person living with dementia, as well as the experiences of the family care partners who attended each session. Within the multiple-case study design, each younger person living with dementia acted as an individual case study and a cross-case sensory analysis was used to build thematic observations underpinning ‘in the moment’ experiences which were seen to exist across cases. The two methods underpinning this study were video-based observation and video-elicitation interviews. In practice, this meant that the first author situated herself as a participant-observer [51] in each of the Music in Mind sessions and placed two cameras on tripods and operated a third from her hand in the seat within the circle. For the video-elicitation interviews, the first author and two members of the wider research team (JK and CS) identified video clips each of around 5–10 min in length from the video recording of the Music in Mind sessions to show to participants as part of video-elicitation interviews. These interviews lasted between 60–120 min and took place within participants’ homes. The three couples took part in joint interviews, whilst an individual (Mary) took part in an individual interview. One person living with dementia (Sam) chose not to take part in an interview but consented to being observed and videoed in the sessions. In total, 42 h of video data were collected within the context of the Music in Mind sessions. 

To answer the new research question, the authorship of this paper was comprised of the lead researcher in the primary study (RD) and two members of the original study team (CS and JK), Manchester Camerata’s most recent community engagement lead for Music in Mind (LH), a musician and specialist in improvisation studies (DHJM), and the lead author of *Care Aesthetics: For Artful Care and Careful Art* [43], whose work is cited several times in this article (JT). To arrive at a secondary data analysis of the primary data collection, we followed a four-step process. First, the findings of the previously published research conducted by RD, JK, and CS were read by the whole authorship to become familiar with the topic area and subject matter. Second, these study findings were discussed and key observations on the data mapped by all members of the interdisciplinary authorship through a series of consensus meetings. Third, we applied Pink’s [52] approach to sensory analysis to the new coding frame to develop emergent themes and highlighted the ‘caring moments’ and embodied musical experiences/engagement with musical-making in this process. Fourth, the concepts of musical improvisation and care aesthetics were then applied to the emergent themes to provide new insights that addressed the reformulated research question. As we will shortly address, this resulted in the identification of the following two key themes: ‘sensing connection’ and ‘curating a creative identity’. Selected edited stills from these video clips will be shared to illustrate these themes and provide additional sensory depth.

### 2.4. Ethical Approval

Ethical approval for this study was granted by the UK’s Social Care Research Ethics Committee (Ref: 16/IEC08/0049).

## 3. Findings

### 3.1. Theme 1: Sensing Connection

This theme highlights the way in which music-making enabled each person living with dementia to feel connected to other members of the group and centres on the relational aspects of ‘in the moment’ experience. Sharing in a unifying experience enabled the person living with dementia to make meaningful connections with other group members, whether it was their spouse, other group members, or the Music in Mind practitioners. This sense of connection allowed the person living with dementia to feel more confident in their musical contributions, feeling more relaxed in the company of the other group members as the sessions went on. The feeling of being connected through group music-making enabled group members to connect physically through touch and enhanced eye contact. Sensing connection is underpinned by the following two subthemes: (1) circles of care and (2) sharing multisensory experiences. 

#### 3.1.1. Circles of Care

The arrangement of the music-making space was pivotal to group members being able to see and respond to the physical and musical improvisations of each other. The instruments were always placed on a table in the centre of a circle of chairs, with the Music in Mind practitioners moving about the space to encourage engagement from each member of the group. Interestingly, each member of the group chose to sit in the same seat each week, and when members of the group were away, these chairs were left empty by the group—suggesting a felt cherishing of the presence of each group member, even in their absence. 

People’s bodies were seen to synchronise both with the pulse of the music and the movements of others. This was most obviously observed when individuals swayed from side to side with the music, their bodies almost ‘marking’ the pulse generated by the group. Synchronisation was not only experienced by single group members, but rather the experience extended to others in the shared space, meaning that there were instances where three or four group members would be moving their bodies in synchrony with each other. This subconscious kinetic entrainment could have enabled the people living with dementia in the group to feel more connected to other group members because their experiences were in synchrony. While this was not recognised directly within the sessions by the group members, there were instances where this kinetic entrainment was acknowledged during video elicitation interviews, as this quote from a family carer highlights—*‘Look we’re all swaying… Even you [RD]!’*

The other means through which the bodies of the people living with dementia synchronised with the bodies of others was through mirroring. There were multiple instances across sessions where group members mirrored gestures, facial expressions, and musical phrases of other group members, the musicians, and the participant-observer. For example, during one warm up activity, group members mirrored Mary’s gestures after her requests to sing a song to which she improvised simple movements (see Figure 1). 

Participants were also able to mirror musical phrases, repeating back melodies or rhythms offered up by the project practitioners or other group members. Although there were no members of the group who had received formal music training, participants were able to listen to the phrases that were offered by other group members and repeat them back. Call and response activities were a central tool used by the Music in Mind practitioners to engage with the group and to facilitate moment-by-moment connections. This created a sense of cohesion, signalling that the person living with dementia was actively listening to the contributions of others, and responding through mimicry.

The circle of care that was created through the shared musical experience also extended beyond the group. The cleaner of the community centre, who had never seen the group in progress, would always set the room up before the session with the circle of chairs. The staff within the community centre could be seen swaying from side to side through the frosted glass windows of their office. The group were also invited to join the weekly community café that happened after the sessions had finished—with their music providing the backdrop for the community chefs, and the smells of the food cooking always piquing the interest of the group. This led to the group beginning to bring food to share with each other—a cherry cake baked by Richard and Jolof rice cooked by Mary. This became a shared ritual at the end of the sessions, sitting and eating together before leaving the group and returning to other activities. 

The group also expressed a sense of care to their individual communities through the sessions. During the 15-week programme, there were two significant events that deeply impacted the members of the group—the Manchester Arena bombing [22 May 2017] and the Grenfell Tower Fire [14 June 2017]. The group mutually shared ways they could give support to the victims of both tragedies and offered support to one another and they also asked the Music in Mind practitioners if they could dedicate their music-making to the people who had been affected. 

Overall, this subtheme showcases the significance of group music-making that happens within a circle. The opportunity to see, hear, and experience music-making together as a group led to greater synchrony and a sense of connection between group members. This was experienced whether a person was living with dementia, a family carer, musician, or participant-observer—highlighting care as multidirectional and not always simply dyadic. Care flowed not only from carers to the people living with dementia, but between people living with dementia, musicians, and the extended community who were using the space. 

#### 3.1.2. Sharing Multisensory Experiences

Improvising together created opportunities for shared multisensory experiences between people living with dementia, family carers, the musicians, and the lead researcher (RD). Each element of the shared music-making process was sensory and embodied, with very few verbal instructions presented by the musicians. Some members of the group were able to sustain musical improvisations for far longer than a verbal conversation. For example, Henry often lost his trail of thought when trying to express himself verbally, but when he was afforded the opportunity to express himself musically, he was able to use his instrument to improvise, playing loudly and confidently, seemingly unafraid of forgetting his words. Furthermore, Henry used his instrument to convey melodies from songs that he knew but could not remember the title of, as this excerpt from an interview with a Music in Mind practitioner illustrates: 


*I knew straight away what he was thinking [in reference to melody Henry has played on the glockenspiel] and I held off saying but I knew exactly what it was ‘cos he’d just done [sings Heidi Hi melody] Heidi Hi, I just knew, I just knew what it was ‘cos it did the same to me ‘cos that’s what I thought the moment he did it.*


These moments of embodied connection between the person living with dementia and other group members, facilitated through musical improvisation, allowed them to develop new relationships or enhance the relationships they already had, such as the relationship between them and their spouse. The quote below highlights Mary’s view that musical improvisation made for a unifying experience: 


*You feel you are coordinating. It’s a communication. You can see all of us when somebody plays this one, plays another one. That is communication. Accepting in the music that we are one because we want to make something good out of what we are doing.*


The shared music-making within the circle provided many opportunities for people living with dementia to be seen and heard through their musical contributions. Many of the group found their musical ‘voice’ through the weeks, finding a sense of confidence through knowing that their contributions were being acknowledged and responded to by the Music in Mind practitioners and other members of the group. This was exemplified especially through the use of a particularly significant piece of music for one member of the group. ‘Da Na Se’ was introduced into the group by Esther in week 5 of the programme. Between sessions 4 and 5, Nicola had researched traditional Ghanaian music after having learned about Esther and Phillip’s cultural heritage. In session 5, she sang them a song called *Nanuma*, which neither Esther nor Phillip appeared to recognise. Esther found this funny, and both Esther and Nicola laughed about the fact that the song was apparently Ghanaian, but Esther had never heard it. This led Esther to share a Ghanaian song of her own with the group: ‘Da Na Se’. Before teaching us the song, Esther told us that it was one of Phillip’s favourite pieces of music and that he had a strong spiritual connection to it. Esther taught the song to the group, using a call and response method with Esther singing the melody, and the rest of the group repeating it back to her. As soon as the group had learned the song, it was performed for Phillip. Phillip’s face lit up on hearing the song performed by the group. The words were difficult for the group to grasp, being in *Twi* (a dialect of the Akan language of Ghana), but the strength of the simple, repeated melody resonated with Phillip, with his face beaming, as he sat forward in his seat.

‘Da Na Se’ became an integral part of each Music in Mind session from that moment forward. Group members were able to observe the significance of the song for Phillip and Esther, and also seemed to enjoy the atmosphere that the song created within the space. A number of the group members discussed the song within the video-elicitation interviews, as the excerpt below highlights: 


*Richard: That song, I don’t know yeah, yeah what sort of atmosphere, well it’s almost sort of hymnal, that’s what I would say [agreement from Carol] and I don’t know what its background is, is it a Christian song I take it?*

*Researcher: Yeah, it means something like thanks be to God, it’s something very simple.*

*Richard: yeah ‘cos it is a song of praise really wasn’t it?*

*Carol: But we all liked it.*

*Richard: Yeah, the tune’s lovely and it’s the sort of tune where you can make up lots of different parts to and the melody sort of carries, so I think it’s lovely.*


Interestingly, although the song started off as being significant for only Phillip and Esther, the rest of the group members adopted this piece of music into their own personal repertoires. There was a sense of musical cohesiveness when group members came together to sing the song week by week. During the final week, the group sang the song for 10 min, almost refusing to move on from the singing of the song that had now become personally significant to each member of the group. 

Overall, this subtheme showcases the importance of the multisensory nature of improvised music-making, and the shift in atmospheres that were created during different fleeting and more extended moments. The co-creation of these different multisensory experiences was important within the space, with group members recognising the significance of different musical preferences and intersections with each person’s cultural, social, and spiritual life story. 

### 3.2. Theme 2: Creative Self-Expression

This theme highlights the ways in which musical improvisation afforded opportunities for people living with dementia to showcase their creative skills, their playfulness, and reveal elements of their self-identity through their improvisations and embodied expressions. The theme is underpinned by the following two subthemes: (1) musical identities and (2) playful performances. 

#### 3.2.1. Musical Identities

People living with dementia were able to showcase elements of their life stories through the music-making experience. As Music in Mind is based on principles of making music ‘in the moment’, rather than explicitly being a reminiscence-based activity, practitioners respond to what is offered up through improvisation rather than having fixed repertoire, songs, or pieces that are performed with the group. As a result, elements of individuals’ life stories can become revealed through and in spontaneous music-making, as aspects of their wider lives, personalities, and interests are offered up within the space.

Different types of accompaniment or melodies resonated with different individuals in the group. Such preferences were outwardly displayed both verbally and non-verbally (e.g., cheering, sitting forward in their seat, dancing). Individuals showed both a preference for certain musical genres (often familiar songs) but also different musical elements (harmony, rhythm, and melody). There were a range of different stylistic features that group members resonated with. For example, Mary enjoyed melodies which were in a major key with simple chord progressions, perhaps reflective of many of the gospel songs that were an integral part of her life, whereas Henry really appreciated rhythms and melodies from Cuban music, as well as Rock ‘n’ Roll.

The quotes below, taken from participant interviews, demonstrate the different musical preferences of the group members both in terms of the music they liked within the session, and their preferences outside of the session: 


*Mary: We pray before they [grandchildren] go to school and they bring up [sings] ‘Oh when the saints go marching on’ which is my favourite.*



*****



*Jenny: you’ve always loved different music. You like your heavy rock didn’t you?*

*Henry: Yeah it was a bit.*

*Jenny: That bloody Kerrang [radio station] stuff that was never, but yeah, we’ve always liked music haven’t we?*



*****



*Barbara: They say that we’re all rhythmic, I mean, that’s we’ve got predominantly one or the other, rhythmic beings, harmonic beings, I mean each of these musical parameters define any individual, I think.*


Improvising together not only provided opportunities to reveal elements of individual life stories but was also shown to support the development of new musical preferences or interests. One of the key principles of Music in Mind is choice, and this was reflected through the ways in which group members interacted with the available percussion instruments across the duration of the programme. In the earlier weeks, while group members were learning the principles of Music in Mind, practitioners would ask group members to choose their instruments, often signalling towards the instrument table as a means of inviting people to choose for themselves. Participants would wait for a member of the group to go to the table and choose, before gaining the confidence to choose an instrument for themselves. However, in the later weeks of the programme, the group members had grown in confidence to make musical choices. This, combined with the values of the facilitated musical space in which every contribution was welcomed, allowed participants to be more confident in their instrument selections. They no longer waited for the instruction to select an instrument but acted with greater spontaneity. One example of this was when Carol and Richard joined the session part way through the ‘Hello Song’. Before even sitting down or taking off her coat, Carol walked with assertion to the instrument table upon entering the room, selected a maraca, then moved rhythmically towards her seat in the circle.

From very early on in the programme it was evident that there were particular instruments that each group member was drawn to. Group members tended to return to the same instruments time and time again, unless encouraged to try something different by the project practitioners. This perhaps showed that each individual had a fondness for particular sonic timbres (e.g., tonal qualities which differentiate the sounds of a tambourine and a woodblock). Group members were also aware of the instrument preferences of others, without these preferences having to be verbalised by the person living with dementia, with many group members mentioning the preferences of others within research interviews as the excerpt below highlights: 


*Richard: And people certainly had their favourites in terms of instruments [agreement from Carol] Henry with the glockenspiel [Carol laughs] and Mary liked the pluck guitar thing [lyre].*


Overall, through the lens of care aesthetics, this could be interpreted as each individual’s identity being validated through their aesthetic choices—for people living with dementia, instead of being viewed through the lens of their diagnosis, they are viewed as a person with different preferences; an awareness of pleasure both in the shape, feel, and sound of musical objects; and as somebody who holds the confidence to improvise with others in a shared music-making space. People living with dementia were thus given a greater degree of caring agency ‘in the moment’, showing awareness of the preferences of others as well as developing an aesthetic connection with the specific musical instruments they chose to play. 

#### 3.2.2. Playful Performances

Through the sessions, people living with dementia were observed to create stories and contribute spontaneously through expressions of the body (as well as sounding using instruments), embodying different emotions and taking on the personas of different characters. They used a range of facial expressions and exaggerated gesture to convey comedy and emotion within their improvising, often using instruments as a ‘prop’ or comedic device. Many of the improvisations created by participants living with dementia brought about humour, opening up opportunities for playfulness between group members. The choreographic and theatrical elements of an individual’s contribution were often driven by embodied musical gesture related to or emerging from interaction with instruments, with group members incorporating kinetic and tactile expressions into their performances, producing exaggerated and often comedic gestures. 

For example, in one exchange when Carol was playing a snare drum, her husband asked a question and she turned towards him and gestured ‘no’ towards him by moving the drum stick from side to side and scrunching her face in a ‘mock’ scolding of Richard (see Figure 2). 

In this interaction, Carol’s contribution to the improvisation expanded from instrument-centred sounding into spontaneous theatricality, through dramatic expressions of the body. She responded to her husband’s provocation by expanding into a full-bodied expression, which also reflected her social relationship with her husband, making it visible within the space. Improvising together with instruments in a playful way facilitated this extension, allowing social and emotional domains (as well as musical) to become activated.

Another member of the group who was particularly playful in his improvisations was Sam. Sam was a member of the group who created more ‘theatrical’ performances within the musical space, adding exaggerated gesture and facial expressions into his instrumental improvisations in order to convey a story. One example of this was captured in the lead authors’ observational notes from a session: 


*During one Music in Mind session, Sam initiated a musical conversation with me using a tambourine. Sam would shake his tambourine in my direction, seeking eye contact with me. I shook my tambourine in his direction as a direct response to his musical initiation. Sam began to vocalise alongside his tambourine playing—gasping and wailing, as if we were using the tambourines to play fight with one another. I joined in with this vocalisation, and our joint performance made the other group members laugh and smile. Barbara improvised an accompaniment alongside our musical ‘fight’ which was in the minor key, using a number of tremolandos (a wavering musical tone) to build the tension and drama to our performance. In this moment, the focus of the whole room was on Sam and I, with everyone laughing along with us in this moment of musical comedy. Interestingly, I was so swept up in this moment that I did not change the view of the camera I was operating to be able to film Sam’s performance in this moment. Capturing the reactions became secondary to our shared experience ‘in the moment’.*


Sam’s ability to add ‘drama’ to his improvisation placed him in the spotlight, affording him a position to express himself non-verbally through spontaneous play. These playful interactions, initiated by Sam and responded to by the Music in Mind practitioners, enabled him to become more relaxed within the sessions and interact with other members of the group. 

Overall, these examples show how the group improvisation provided an opportunity to create playful exchanges and seek moments of comedy, enabling the people living with dementia to express themselves through the performing of their own stories. Such holistic performance, across domains of sound as well as body movement and emotional expression, were encouraged in the Music in Mind programme through the facilitation of an improvised musical space which upheld the values that every contribution was recognised and welcomed. 

## 4. Discussion

This article presents an exploration of an improvisation-based group music programme delivered by Manchester Camerata (Music in Mind), for people living with young onset dementia through the lenses of care aesthetics and musical improvisation. The re-examination of the original study [44] with experts in improvisation studies (DHJM) and care aesthetics (JT) led to the development of two central themes—‘sensing connection’ and ‘creative self-expression’—which shed light on the impacts that group music-making sessions can have on participants’ sense of connectedness, relational dynamics, and opportunities for creative expression while living with a diagnosis of young onset dementia. The following sections will discuss the central findings and subthemes alongside wider literature relating to the experience of young onset dementia, care aesthetics, and musical improvisation. 

An aesthetic choice was observed in the intentional design of the music-making space, which was seen to foster a sense of connection and belonging among participants. While this has been documented in the architectural design of care spaces [53], including how ‘design and spatiality’ create ‘embodied practices, atmospheres, and affects’ [54], here the iterative co-creation of space developed an emergent aesthetics of care in play for the group. The arrangement of chairs in a circle, the incorporation of culturally significant songs, and the recognition of each member of the group through their contributions to the improvisation (rather than their diagnosis) all generated a caring environment which valued the unique identities and preferences of the group members. In essence, care was reflected in the relational circles created within the physical circle, with the music co-created between the members acting as a vector for connection, as well as a site of non-physical touch. The circle has a pedagogical ethic, of inclusion and collaboration, a care ethic in the way that all are invited to see and be seen by each other, and then a care aesthetic in that the quality of the shape effects the sensation of the experiences that take place within it. The facilitated space was also shown to set up a value-system in which everything was welcomed, such that during interactive improvisation, participants living with dementia were not necessarily ‘judged’ or limited by the cognitive challenges that may have affected them in other parts of their daily lives [25]. This reflected, spatially, the inherent accessibility [29] and ethic of acceptance [30,55], which underpin group improvisational practice. 

For people living with young onset dementia, this opportunity to be welcomed into a judgement-free space is especially significant given the lack of appropriate services catering for people from a younger demographic [18], a situation that has been recognised in the United Kingdom for some considerable time [56]. Over a decade ago, Davies-Quarrell et al. [57] outlined an evaluation of the ACE Club, a long-standing specialist support service for younger people living with dementia in the community, which was conducted by the group members themselves. The evaluation questions that the ACE Club members set themselves was to explore what made their service meaningful and helpful (to themselves) and why, to put it bluntly, they kept returning week after week. Whilst the ACE Club reported many benefits to their regular meetings, including a belief that friendship helped the group to keep living well with their dementia, by far the most important factor was a sense of co-created group cohesion. As these authors reported, this group cohesion provided ‘a sense of security for its membership where “permission” to be vulnerable within a supportive environment was essential to human growth’. In many ways, the regular routine and performative activities of the Music in Mind sessions also created a sense of security and belonging to its membership, a finding that chimes with the earlier work of the ACE club members and the sense of fun and enjoyment that was so crucial in keeping people living with young onset dementia returning to the same space and activities. Arguably, leaving room for exploring and evaluating the fun and enjoyment of everyday activities would seem an important future step forward in dementia studies, no matter what the age of participants. 

Engaging with group-based musical improvisation also saw connections between people living with dementia and their family members grow, as well as the development of new friendships between group members. Both care aesthetics and musical improvisation highlight the importance of relational dynamics and interpersonal connections in caregiving contexts [36,58,59,60], and several of the techniques used by Music in Mind practitioners (i.e., call and response) encouraged the group to pay greater attention to small movements, sounds, and musical contributions. This was showcased through mirroring and bodily synchrony, where subtle movements were adopted (either intentionally or automatically) by group members through the shared music-making experience. To mimic, mirror, copy, or synchronise can be an act of endorsement or recognition—‘I’m seeing you, I’m hearing you’—it is a careful act of attention to another person or persons. Aesthetically and relationally in musical improvisation, sounding together can cultivate a sense of cohesiveness, a common ground, and a mutual support in which all parties contribute to and attune within a shared soundscape. Meeting the ‘other’ through attentive co-sounding, or co-moving, represents common practice within therapeutic improvisational methodologies in music, as well as dance [32,61,62]. Where ‘attentiveness’ is a key component of *caring about* the other proposed in the care ethics literature, most famously by Joan Tronto [63], the attentiveness here has a full embodied or sensory quality. It has a relational aspect not only through a carer attending to a person cared for, but more dynamically through the circle of care that creates a series of attentive practices between participants. Situated within the highly relational, co-creative space of group improvising, synchronicity becomes an engine of the aesthetic experience.

Playful and joyful musical improvisation created opportunities for an exciting and stimulating musical environment in each session. Many of the group’s improvisations led to humorous and playful exchanges between group members, Music in Mind practitioners, and the in situ researcher (RD). Playful approaches when improvising in dementia contexts have been explored more purposefully in recent years, including research into relational clowning [64,65], with practitioners and researchers suggesting that playfulness in the everyday lives of people living with dementia can enhance connection and provide ‘meaningful moments’ for people living with dementia and their care partners, as well as professionals leading and facilitating improvisation exercises [66,67]. Recognising group improvising as a ‘particularly sophisticated type of socially mediated artistic collaborative endeavour’ [29], with the multisensory, embodied playfulness across instruments and body in Music in Mind sessions represents a site of social, as much as creative, significance. 

Furthermore, there are a number of different types of humour that have been defined, but the one most associated with the displays of humour observed during musical performances is affiliative humour, which is humour used to amuse others and strengthen social relationships [68]. There is very little focus placed on humour within the dementia studies literature, even though many people living with dementia report the importance of playfulness and humour in their lived experience of dementia [69]. Humour has been explored as a way to be a salient and a meaningful way for people living with dementia to cope positively with their diagnosis [70]. As noted in the introduction, a diagnosis of dementia in mid-life can have serious and disruptive consequences for a person’s life choices and decision-making abilities, and thus their ability to cope positively with a diagnosis. Thus, musical improvisation in a group context may provide opportunities for finding meaningful, playful experiences that are co-created and held ‘in the moment’, thereby promoting holistic wellbeing and emotional resilience. Music in Mind is not an example that presents a solitary aesthetic experience, but crucially one that becomes shared between group members, whether musicians, carers, or people living with dementia. Humour is a vital register of this experience and illustrates how attending to each other creates a shared, meaningful ‘relational aesthetic’ [71]. Being ‘in the moment’, responsive improvisation, and an openness to an aesthetics of care thus become three intersecting frames for giving value to the collective and rich sensory life within a particular group. 

### 4.1. Implications and Future Directions

Music in Mind is a time-limited programme that supports the use of musical improvisation in the context of dementia. There is growing recognition of the ways in which everyday acts of creativity can support individual flourishing and individual wellbeing [72], and there is scope to explore the everyday ways in which people living with dementia and their care partners ‘improvise’ through their dementia journey in different embodied ways [46]. The use of music in a young onset dementia context is significantly underexplored compared to typical onset dementias, with the majority of research taking place within care home contexts [73]. While there are many positive benefits of engaging with group improvisatory music-making for people living with dementia more generally, people living with a younger onset are a key group who could benefit from activities of such type because it creates opportunities for emotional engagement and social connection with those experiencing dementia in mid-life—whether this is the person with dementia or their family carer. 

Using a lens of care aesthetics is also significant in this area of research and practice. Care aesthetics underscores the significance of recognising and valuing the unique identities and preferences of people living with dementia, and values the ‘in the moment’, relational, and embodied experiences of everyone involved in music-making opportunities. By honouring participants’ musical preferences, choices, and creative expressions, such improvised music-making sessions have the potential to validate personal identities and agency, fostering a sense of empowerment and self-worth through shared creative experiences. There is scope here to develop rigorous mixed method approaches which capitalise on data collected by video, which could complement the methods used in this study to explore ‘in the moment’ musical experiences on a greater scale in the future.

### 4.2. Study Limitations

It must be acknowledged that while the goal of case study research is not to build generalisable findings, the sample size in this study was set within a specific musical context and group (Music in Mind). The in-depth nature of this study allowed for the exploration of ‘in the moment’ embodied and sensory experiences, but it is not clear whether these findings would extend beyond the study participants. Therefore, clearly there is scope to develop these themes further in the context of a larger sample and different musical or improvisational context in the future.

Secondly, due to the focus of this research being ‘in the moment’, there was no formal follow-up with participants to examine the longer-lasting impacts of having taken part in the Music in Mind sessions. The sense of connection that had been felt by group members meant that the final session was both a celebration and a ‘goodbye’ for many of the group members who no longer had the weekly opportunity to meet with the friends they had made in the group. Following-up with participants later down the line may have given more insight into the lasting benefits or challenges experienced by participants as a result of no longer having weekly sessions to attend.

## 5. Conclusions

Musical improvisation has the potential to support the psychological, social, and spiritual wellbeing of people living with young onset dementia, at a time when there are many added stressors associated with diagnosis in mid-life. Applying a lens of care aesthetics and musical improvisation, it is possible to observe the micro-level experiences of people living with dementia and their family carers—learning what it means to be ‘in the moment’ from an individual and relational level. There are still few dedicated music programmes which actively focus on young onset dementia, but dedicated focus on developing and delivering such programmes could support people living with dementia to feel more connected to themselves, to other people, and the wider communities in which they live. 

## Figures and Tables

**Figure 1 ijerph-21-00972-f001:**
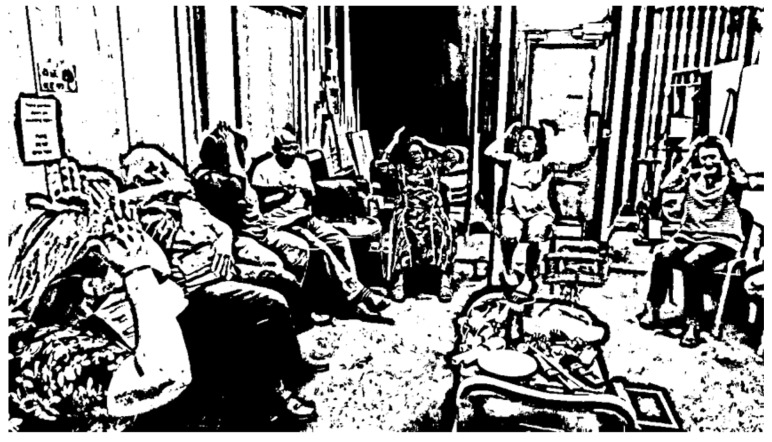
Group members mirror Mary’s improvised movements.

**Figure 2 ijerph-21-00972-f002:**
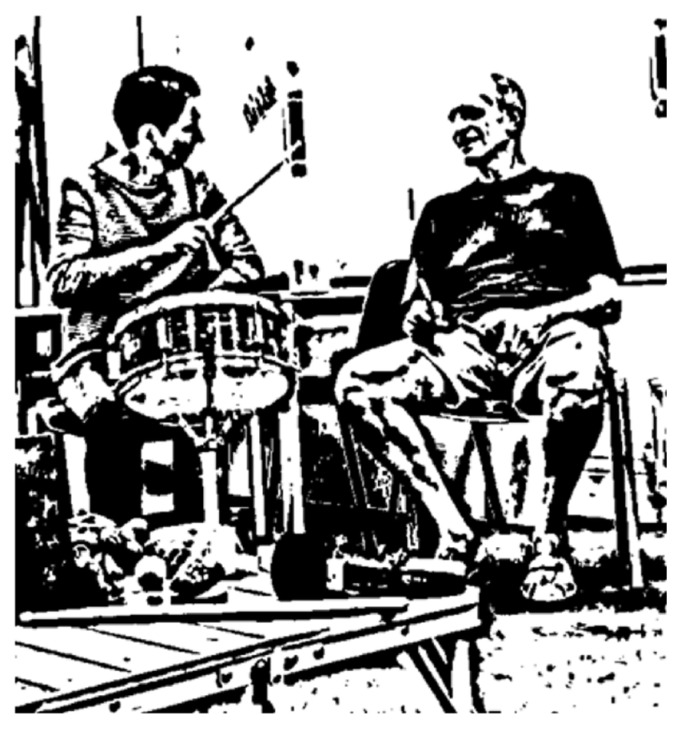
Carol playfully scolds her husband.

**Table 1 ijerph-21-00972-t001:** Participant characteristics.

	Pseudonym	Diagnosis	Gender	Age	No. of Sessions Attended *
Couple 1	Henry	Posterior Cortical Atrophy	Male	62	12
	Jenny	-	Female	59	12
Couple 2	Phillip	Lewy Body Dementia	Male	59	11
	Esther	-	Female	57	11
Couple 3	Carol	Young Onset Alzheimer’s Disease	Female	59	11
	Richard	-	Male	70	11
	Mary	Young Onset Alzheimer’s Disease	Female	60	13
	Sam	Young Onset Alzheimer’s Disease	Male	60	12

* Out of a maximum of 15 sessions.

## Data Availability

No new data were created or analysed in this study. Data sharing is not applicable to this article.

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
