# Peer review of "Creating and Reliving the Moment: Using Musical Improvisation and Care Aesthetics as a Lens of Connection and Self-Expression for Younger People Living with Dementia"

_ijerph, 2024, doi:10.3390/ijerph21080972_

Round 1
Reviewer 1 Report
Comments and Suggestions for Authors
I would like to congratulate the authors on this excellent analysis and write-up, and thank the editor for the opportunity to review, ‘Creating and Reliving the Moment: Using musical improvisation and care aesthetics as a lens of connection and self-expression for younger people living with dementia’.
The manuscript is clear, and the findings incredibly valuable for promoting the use of improvisatory music with and for people living with young-onset dementia. On an anecdotal level, I could certainly relate to the themes extrapolated from the data based on my own experiences of conducting music research with people living with younger-onset dementias.
A few minor points which I hope are helpful:
Some of the themes chimed with work on the With All project, conducted by Hannah Zeilig, Julian West, and Millie van der Byl Williams, which may be useful to incorporate within the introduction: https://www.emerald.com/insight/content/doi/10.1108/QAOA-02-2018-0008/full/html
In the Improvisatory Performance section, the authors could raise that improvisation is a great equaliser for diverse groups of people coming together for the first time. It may be important to raise this, as we look to better representation in arts activities for dementia; improvisation removes the need to rely on a shared musical lexicon which can be othering when not shared by all, and indeed this aligns with the Theme 3.1.2, ‘Sharing multisensory experiences’.
I was particularly interested in the notion of care aesthetics, and specifically lines 142-143 which touch on ‘valuing the subtleties of micromovements (etc.)’ which can also be quantified with various software. I wondered if the authors would consider a brief comment (perhaps in the discussion) about how mixed methods approaches could capitalise on data collected via video to provide further insight in this field, perhaps in larger scale group work in this field?
Table 1 may benefit from some minor formatting to highlight the dyads where relevant, plus noting sessions attended were out of a maximum of 15. It would also be helpful for participant ethnicity to be included, as the group appears to be more representative than most music and dementia research which is a strength.
Author Response
Thank you for taking the time to read and review this manuscript. We really appreciate your comments and your interest in this topic area. Our changes in line with your comments are listed below.
Some of the themes chimed with work on the With All project, conducted by Hannah Zeilig, Julian West, and Millie van der Byl Williams, which may be useful to incorporate within the introduction: https://www.emerald.com/insight/content/doi/10.1108/QAOA-02-2018-0008/full/html
We are really inspired by the With All project and the work of Zeilig et al – this work is already cited in the introduction (reference 26). We have expanded on the point to draw focus to the specific conclusions of this work in the following sentence on line 94.
In the Improvisatory Performance section, the authors could raise that improvisation is a great equaliser for diverse groups of people coming together for the first time. It may be important to raise this, as we look to better representation in arts activities for dementia; improvisation removes the need to rely on a shared musical lexicon which can be othering when not shared by all, and indeed this aligns with the Theme 3.1.2, ‘Sharing multisensory experiences’.
Thank you for this comment – we have added a sentence which highlights the equalising of the space that is made possible through improvisation lines 93-96.
I was particularly interested in the notion of care aesthetics, and specifically lines 142-143 which touch on ‘valuing the subtleties of micromovements (etc.)’ which can also be quantified with various software. I wondered if the authors would consider a brief comment (perhaps in the discussion) about how mixed methods approaches could capitalise on data collected via video to provide further insight in this field, perhaps in larger scale group work in this field?
Thank you for this suggestion – we agree that rigourous mixed-methods approaches which do not centre on ‘symptoms’ alone are really necessary within this area of research. We have added a sentence to the future directions section highlighting this on lines 691-694.
Table 1 may benefit from some minor formatting to highlight the dyads where relevant, plus noting sessions attended were out of a maximum of 15. It would also be helpful for participant ethnicity to be included, as the group appears to be more representative than most music and dementia research which is a strength.
Thank you for this comment – we agree that the table needed to be clearer. We have separated out the couples in the table and hope this makes it clearer. We have not included participant ethnicities within the table as this was something we were not able to formally collect (ethical requirement) but the cultural heritages of the group became apparent through the music making and social engagement across the sessions. We wouldn’t want to put this information into the table as we do not have a formal record of self-reported ethnicity and wouldn’t want to make any assumptions on this.
Reviewer 2 Report
Comments and Suggestions for Authors
Dear authors,
You have presented a beautiful approach to working with young-onset dementia. I am calling your attention to several thematic concerns that if addressed will strengthen your paper: 1) distinguishing between improvisation and performance is essential when discussing music therapy and/or the therapeutic affordances of improvisation. 2) your discussion would be enhanced within a framework of group dynamics and its tenets such as attunement, altruism, and universality. I have expounded on these suggestions along with providing suggestions on content, grammar, and sentence structure in the attached document.

Comments on the Quality of English Languageincluded in word doc
Author Response
Thank you for your detailed review of our manuscript - we have attached a document with our responses to your comments and where changes have been made are highlighted in red in the revised manuscript.

Reviewer 3 Report
Comments and Suggestions for Authors
Congratulations on designing this thoughtful study that explores the multifaceted impacts of improvised music-making program on individuals with younger onset dementia. I have some minor comments as below:
Please provide a STROBE checklist for observational studies.
I understand that this may be perceived as a qualitative study, but more could be done to improve the data collection for better understanding on this program. For instance, psychological statuses (anxiety, QOL, loneliness et), functional abilities, or even music related outcomes such as musical engagement should be measured via validated instruments before and after the program.
Additionally, this study reported only those who received the intervention, but there was no control group to further determine the effects of the program on the participants. The small sample size, albeit mentioned in the limitations, was indeed restricting the generalizability of the findings given the fact that the cohort comprised both patients of different diagnosis and caregivers.
Comments on the Quality of English Language
N.A.
Author Response
Thank you so much for taking the time to review this manuscript. We really appreciate your thoughts on the manuscript and your perceptions of its value. Please note our responses below.
Please provide a STROBE checklist for observational studies.
Thank you for making us aware of the STROBE checklist. We have examined the checklists available online and we do not believe the STROBE checklist is applicable t in the context of our qualitative case-study design. We cannot present quantitative data as it was not within the scope of the aims/objectives/design of the study.
I understand that this may be perceived as a qualitative study, but more could be done to improve the data collection for better understanding on this program. For instance, psychological statuses (anxiety, QOL, loneliness et), functional abilities, or even music related outcomes such as musical engagement should be measured via validated instruments before and after the program.
Thank you for your point relating to psychological statuses, functional abilities, and musical engagement. The aims of this study (as presented in the paper) did not intend to find any differences in psychological statuses, functional abilities or musical engagement, but rather understand the ‘in the moment’ experiences of people with dementia. These outcomes have been presented extensively within the wider music and dementia literature but was not our focus in this study. This is data we cannot present as it was not collected in this study.
Additionally, this study reported only those who received the intervention, but there was no control group to further determine the effects of the program on the participants. The small sample size, albeit mentioned in the limitations, was indeed restricting the generalizability of the findings given the fact that the cohort comprised both patients of different diagnosis and caregivers.
Thank you for your point regarding a control group and lack of possible generalizability. With case study research, the point is not to produce generalizable findings and we have highlighted the limitations in text. There was no control group in this study when the data was collected. The focus of this study was not to produce findings of difference (before/after; music/control) but to explore ‘in the moment’ experiences in this given context. We feel that we have justified the use of these methods sufficiently in the text and cannot provide data from a control group as it didn’t exist within the study design.